# Identification of Sulfate Transporter Genes in *Broussonetia papyrifera* and Analysis of Their Functions in Regulating Selenium Metabolism

**DOI:** 10.3390/plants14192995

**Published:** 2025-09-27

**Authors:** Yaobing Chen, Nuo Wang, Chengxu Qian, Weiwei Zhang, Feng Xu, Qijian Wang, Yongling Liao

**Affiliations:** 1School of Forestry and Horticulture, Hubei Minzu University, Enshi 445000, China; 13971878851@163.com; 2College of Horticulture and Gardening, Yangtze University, Jingzhou 434025, China; 2022710851@yangtzeu.edu.cn (N.W.); 15972821412@163.com (C.Q.); wqjjeans@163.com (Q.W.); liaoyongling@yangtzeu.edu.cn (Y.L.); 3Hubei Key Laboratory of Spices & Horticultural Plant Germplasm Innovation & Utilization, Yangtze University, Jingzhou 434025, China

**Keywords:** *Broussonetia papyifera*, sulfate transporter, selenium accumulation, functional analysis

## Abstract

*Broussonetia papyrifera* has strong adaptability and exhibits a strong ability to accumulate selenium. Its leaves are rich in crude protein, amino acids, and minerals, making them high-quality feed materials. To improve the selenium-enriched ability of *B. papyrifera* and promote the development of selenium-enriched agricultural products, we screened and identified the sulfate transporters associated with selenium absorption in *B. papyrifera*. By treating the leaves of *B. papyrifera* with different concentrations of sodium selenate and analyzing the correlation between gene expression and selenium content, we identified *BpSULTR3;1* and *BpSULTR3;4*, which may be involved in selenium absorption and transport in *B. papyrifera*. We further validated the functions of *BpSULTR3;1* and *BpSULTR3;4* through transgenic experiments in *Arabidopsis thaliana*. The results showed that overexpressing *BpSULTR3;1* significantly increased the total selenium content in *A. thaliana*, up to 2.31 times, and also increased the contents of three forms of organic selenium (SeCys_2_, MeSeCys and SeMet) in transgenic *A. thaliana*. These findings provide solid theoretical support for improving *B. papyrifera*’s selenium enrichment ability through genetic improvement.

## 1. Introduction

*Broussonetia papyrifera*, a deciduous tree or shrub in the Moraceae family, shows wide adaptability, drought resistance, and strong tolerance to humidity. As a prominent eco-economic species, it has been selected as the primary candidate for afforestation plans in China. This tree is a rich source of amino acids and proteins, making it a high-quality raw material for feed. Using *B. papyrifera* leaves as silage can reduce rumen biohydrogenation and increase polyunsaturated fatty acids in milk. The leaves have high levels of crude protein, crude ash, crude fat, phosphorus, and suitable levels of crude fiber, which can help mitigate China’s protein feed shortage and reduce the dependence on foreign sources [1,2]. The leaves also contain various bioactive compounds, such as flavonoids, phenylpropanoids, and polyphenols, which are often used to treat many diseases. The bark is rich in cellulose, making it a high-quality raw material for papermaking [3,4,5]. Research indicates that *B. papyrifera* contains various trace elements with pharmacological activities, including iron (Fe), manganese (Mn), copper (Cu), and molybdenum (Mo) [3,6]. Fe is essential for hemoglobin synthesis in the human body, while Mo and Mn are important for cancer prevention and treatment. Moreover, *B. papyrifera* is a pioneer tree species with significant ecological value, contributing to heavy metal pollution remediation, harmful gases absorption, and soil erosion prevention [7,8].

Selenium (Se) is an essential trace element for humans and animals, mainly found in the body as selenoproteins that perform crucial biological functions. So far, 25 distinct selenoproteins have been identified in humans, playing vital roles in antioxidant defense, immune regulation, and cancer prevention [9,10]. A lack of selenium can disrupt normal bodily functions, leading to diseases such as Keshan disease and Kaschin-Beck disease. Scientific selenium supplementation has been proven effective in alleviating deficiency-related symptoms. However, about 72% of China’s land area lacks selenium, resulting in inadequate selenium intake per person [11]. To address this issue, using selenium-enriched plants has emerged as an economical, efficient, and safe strategy for dietary selenium supplementation. This is due to plants’ unique ability to convert inorganic selenium into bioavailable organic forms, including selenocysteine, selenomethionine, and methylselenocysteine, which are then incorporated into selenoproteins, enhancing their nutritional accessibility and safety [10,12]. Although selenium has not been classified as an essential element for plants, numerous studies have confirmed its positive effects on the physiological and nutritional indices of various plant species. At appropriate concentrations, selenium can promote plant growth and development, enhance nutritional quality [13,14], and facilitate fruit preservation, improving their storability and transportability [15]. Moreover, adequate selenium exerts beneficial effects on plant resistance to multiple abiotic and biotic stresses, including cold, drought, pathogens, and heavy metals [16,17,18]. In recent years, multiple studies have confirmed that *B. papyrifera* exhibits a strong ability to accumulate selenium [19,20]. Chen et al. found that exogenous Se treatment on *B. papyrifera* significantly increased the content of selenomethionine (the major organic Se form) in its leaves, while also elevating the contents of nutritional substances including soluble sugars, phenolic acids, and flavonoids, thereby improving the quality of *B. papyrifera* [6]. This characteristic not only enhances the content of bioactive substances and nutritional components in its leaves, improving its value as feed, but also enables the diversified utilization of *B. papyrifera* resources. Specifically, animal feed processed using *B. papyrifera* as a plant-derived selenium source can indirectly provide humans with a safe and effective selenium supplementation pathway through the food chain, which holds significant implications for expanding the sources of selenium-enriched feed and improving population selenium nutrition status.

Selenium mainly exists as selenate in soil, and plants primarily absorb selenate through the sulfate transport pathway. This process takes place specifically through active transport that is mediated by sulfate transporters (SULTRs), which facilitate selenate entry into plant cells, modulate selenium uptake, and metabolic processes, representing a crucial step in plant selenium accumulation [21,22]. This is because sulfur and selenium belong to the same main group in the periodic table of elements, exhibiting similar structures and functions, and SULTRs demonstrate a high affinity for these elements. Interestingly, SULTRs have also been found to be involved in the accumulation of phosphorus and phytic acid in plants, such as the *OsSULTR3;3* gene in rice [23,24]. Additionally, studies on the order of selenium and sulfur absorption have shown that when selenates and sulfates coexist in the environment, non-selenium-enriched plants preferentially absorb sulfur, whereas selenium-enriched plants prioritize the absorption of selenium [25,26]. As a result, the Se/S absorption ratio in plants with high selenium accumulation is higher than that of other plants growing under identical conditions [27]. For instance, the selenium hyperaccumulator *Stanleya pinnata* expresses *SpSULTR2;1* at high levels, allowing it to maintain efficient selenium accumulation even in high-sulfur environments [28]. Based on their functional differences and localization, SULTRs in plants are categorized into five groups: SULTR1, SULTR2, SULTR3, SULTR4 and SULTR5 [29]. The high-affinity proteins in group I, including AtSULTR1;1, AtSULTR1;2, and AtSULTR1;3, are primarily involved in selenate uptake and transport. AtSULTR1;1 and AtSULTR1;2 are localized in the root system and facilitate selenate uptake by roots [30]. In contrast, AtSULTR1;3 is mainly distributed in the phloem and catalyzes selenate transport within this tissue [31]. Group II consists of low-affinity sulfate transporters that are widely present in various plant tissues. AtSULTR2;1 is localized in the xylem, leaf phloem, and root peripheral cells, while AtSULTR2;2 is found in the root phloem and leaf vascular bundle sheath cells, together mediating selenate translocation in vascular tissues [32]. The five members of Group III (AtSULTR3;1 to AtSULTR3;5) are localized on chloroplast membranes and facilitate selenate transfer to chloroplasts [33]. Group IV includes low-affinity proteins, such as AtSULTR4;1 and AtSULTR4;2, which may be involved in selenate efflux from vacuoles [34]. Group V comprises AtSULTR5;1 and AtSULTR5;2, which do not participate in selenate transport [35]. The regulation of selenate uptake in plants is primarily mediated by high-affinity sulfate transporters in the roots, involving active transport [36]. Once inside plant cells, selenate is reduced to selenite through the sulfur metabolic pathway and further converted into organic selenium compounds, such as selenocysteine (SeCys) and selenomethionine (SeMet).

SULTRs, key transporters for selenate uptake, play a crucial role in selenium metabolism and accumulation in plants. By investigating how SULTRs mediate selenium enrichment in *B. papyrifera*, we can expand our understanding of the molecular regulatory networks governing plant selenium metabolism. This will also increase the exploitation and utilization value of *B. papyrifera*. Moreover, this research will provide a scientific basis for promoting selenium-enriched plant-based feed.

## 2. Materials and Methods

### 2.1. Plant Material and Selenium Treatments

The *B. papyrifera* material used in the experiment was 1-year-old hybrid seedlings, the variety was ‘Kegou 101’. These seedlings were planted in circular pots with a height of 14 cm and a top diameter of 12 cm. The nutrient substrate was a mixture of peat, red soil, vermiculite, and perlite at a volume ratio of 6:1:1:1. All seedlings were uniformly cultivated in greenhouse of Yangtze University, Hubei (30° 37′ N,112° 07′ E). The growth conditions were set as follows: ambient temperature of 25 °C, relative humidity of 65%, and a photoperiod of 12 h light/12 h dark. Meanwhile, regular water and fertilizer management were conducted throughout the cultivation period. After 40 days of transplanting, plants with uniform growth were selected for selenium treatment. The treatment groups were supplied with 0.2, 0.4, 0.8 mmol/L Na_2_SeO_4_, whereas the control group was provided with water. The leaves were sprayed evenly on both sides using a spraying pot every seven days for a total of 4 times (As a deciduous woody tree, *B. papyrifera* has a longer growth cycle, thicker leaf cuticles and larger biomass compared to herbaceous plants). Five days after the last treatment, the residual sodium selenate in the leaves was washed off with water. Two days later, the leaves were collected, frozen in liquid nitrogen, and stored at −80 °C. Each treatment group had 3 biological replicates, with each replicate consisting of 3 independent plants.

Wild-type *A. thaliana* (ecotype Columbia) was used as the recipient in transgenic experiments. To sterilize the seeds, they were first treated with 75% ethanol for 30 s, then rinsed twice with sterile water. Next, they were then treated with a 2.5–3% sodium hypochlorite solution for 6-8 min, followed by three rinses with sterile water. The sterilized seeds were air-dried on autoclaved filter paper and then sown on pre-prepared MS solid medium and incubated in a plant growth chamber. After 5–7 days, the germinated seedlings were transplanted into a mixed substrate (peat: turfy soil: perlite = 7:2:1) and covered with plastic film for a week to maintain humidity. The plants were then grown in a growth chamber at 22–25 °C with a 16 h light and 8 h dark photoperiod. A total of 15 days after growth, the plants were used for floral dip transformation to generate transgenic lines. T_3_ homozygous transgenic plants were sprayed with 0.4 mmol/L Na_2_SeO_4_ until the solution began to run off the leaf surfaces. This treatment was repeated every 7 days for a total of 3 applications (As a model herbaceous plant, *A. thaliana* has thin leaf cuticles, a short life cycle and high metabolic activity). A total of 4 days after the final treatment, the plants were harvested after a thorough rinse with deionized water and then subjected to subsequent selenium content analysis.

*Nicotiana benthamiana* was used for subcellular localization assays. Prior to use, its seeds were stored in a refrigerator at 4 °C. The seeds were then sown in a nutrient substrate consisting of peat, vermiculite, and perlite at a volume ratio of 7:2:1. Then seedlings were transplanted at the 3–4 true-leaf stage. After transplantation, the plants were covered with plastic film to retain moisture and incubated in a growth chamber. The film was removed one week later, and plants were grown for another 20–25 days before being used in subcellular localization experiments.

### 2.2. Identification of the SULTR Gene Family in B. papyrifera

The genome data of *B. papyrifera* used in this study were obtained from the research by Chen et al. [37]. The AtSULTR protein sequences of the *A. thaliana* were retrieved from the *Arabidopsis* database (TAIR, http://www.arabidopsis.org/ (accessed on 18 April 2024)). These sequences were compared with the genome data of *B. papyrifera* using the local BLAST +2.14.0 program, with an E-value threshold set to 1e-5. Concurrently, the Hidden Markov Model (HMM) seed files for the SULTR family (Pfam ID: PF00916 and PF01740) were obtained from the PFAM database (http://pfam.xfam.org/ (accessed on 24 April 2024)). The homologous sequences in the *B. papyrifera* genome database were then scanned using HMMER 3.0 software. By integrating the search results from BLAST and HMMER, the candidate sequences were validated through the PFAM database and the SMART website (http://smart.embl-heidelberg.de/ (accessed on 25 April 2024)). Sequences containing both the characteristic domains of sulfate transporters and STAS were then filtered, ultimately determining the members of the SULTR gene family.

### 2.3. Bioinformatics Analysis of SULTR Gene Family in B. papyrifera

The physical and chemical properties of the protein were predicted using the ExPASy online tool (http://web.expasy.org/protparam/ (accessed on 3 May 2024)). The TMHMM-2.0 (http://www.cbs.dtu.dk/services/TMHMM/ (accessed on 3 May 2024)) was utilized for transmembrane analysis. Hydrophobicity analysis was performed using the ProtScale platform (http://web.expasy.org/protscale/ (accessed on 11 May 2024)). The prediction of subcellular localization was achieved through the Cell-PLoc website (http://www.csbio.sjtu.edu.cn/bioinf/Cell-PLoc-2/ (accessed on 12 May 2024)). The prediction of cis-acting elements of promoters was based on the PlantCare database (http://bioinformatics.psb.ugent.be/webtools/plantcare/html/ (accessed on 15 May 2024)) and visualized using Tbtools-II v2.326 software [38]. The SOPMA online tool (https://prabi.ibcp.fr/htm/site/web/app.php/ (accessed on 18 May 2024)) was employed to predict the secondary structure of the protein. Finally, the modeling and analysis of the protein’s tertiary structure were completed using the SWISS-MODEL platform (http://swissmodel.org/workspace/ (accessed on 20 May 2024)).

### 2.4. Multiple Sequence Alignment and Phylogenetic Analysis

Multiple sequence alignments were conducted using MEGA11.0 software on the screened and identified SULTRs of *B. papyrifera*. The phylogenetic tree of SULTRs from *B. papyrifera* and *A. thaliana* was constructed utilizing the Maximum Likelihood (ML) method. The bootstrap analysis was performed with 1000 repetitions, and the resulting phylogenetic tree was enhanced and visualized using the iTOL online software (https://itol.embl.de/ (accessed on 23 May 2024)).

### 2.5. RNA Extraction and RT-qPCR Analysis

Total RNA was extracted from plant material using the TaKaRa MiniBEST Plant RNA Extraction Kit (TaKaRa, Beijing, China). The concentration and integrity of the RNA were assessed using a NanoDrop spectrophotometer (Thermo Fisher Scientific, Waltham, MA, USA) and gel electrophoresis. First-strand cDNA synthesis was conducted with the HiScript III 1st Strand cDNA Synthesis Kit (+gDNA wiper) (Vazyme, Nanjing, China). Quantitative primers were designed utilizing Primer Premier 6.0 software, and primer specificity was verified using TBtools software. *AtPP2A* [39] served as the reference gene for *A. thaliana*, while *BpUbiquitin* [6] was used as the reference gene for *B. papyrifera*. The primers were synthesized by Shanghai Bioengineering Co., Ltd., and all primer sequences are presented in Appendix A. The RT-qPCR analysis was conducted using LineGene 9600 Plus system (BioerTechnology, Hangzhou, China). The expression levels of the target genes were quantified using the 2^−ΔΔCt^ method [40].

### 2.6. Subcellular Localization of BpSULTR3;1 and BpSULTR3;4 Proteins

To further investigate the subcellular localization of *BpSULTR3;1* and *BpSULTR3;4* protein, pICH86988-*BpSULTR3;1/BpSULTR3;4*-GFP fusion expression vector was constructed. The recombinant plasmid was transformed into the *Agrobacterium tumefaciens* strain GV3101, and then inoculated into YEP liquid medium supplemented with 50 μg/L kanamycin (Kan) and 20 μg/L rifampicin (Rif). The culture was incubated at 28 °C with shaking at 180 rpm until the optical density at 600 nm (OD_600_) reached 0.5–0.6. Bacterial cells were harvested by centrifugation at 5000 rpm for 10 min at room temperature, resuspended in a working solution composed of 1 mL of 0.5 mol/L MES, 1 mL of 0.5 mol/L MgCl_2_, and 10 μL of 1 mol/L acetosyringone, and then adjusted to a final volume of 50 mL with water. The OD_600_ of the suspension was adjusted to 1.0, and it was incubated in the dark at room temperature for 2 h. Five-week-old healthy *N. benthamiana* plants were utilized for agroinfiltration. Following infiltration, the plants were cultured in the dark for 16 h, after which they were incubated in a light-growth chamber for 2 days. Infiltrated leaf tissues (0.8 cm × 0.8 cm) were excised, mounted on slides, and observed using a laser scanning confocal microscope. The pICH86988-GFP empty vector served as the control.

### 2.7. Construction and Transformation of BpSULTR3;1 and BpSULTR3;4 in A. thaliana

The plant overexpression fusion vector pCYH05252-*BpSULTR3;1/BpSULTR3;4* was constructed and then transformed into the *Agrobacterium tumefaciens* strain GV3101 for primary activation. The transformed agrobacteria were inoculated into YEP liquid medium supplemented with 20 mg/L Rif and 50 mg/L Kan, and then incubated at 28 °C with shaking at 200 rpm for 2 days. For secondary activation, 300 μL of the primary culture was transferred to 7 mL of fresh YEP liquid medium containing the same concentrations of Rif and Kan. The culture was incubated overnight at 28 °C with shaking at 200 rpm until the OD_600_ reached 0.8–1.0. The bacterial cells were harvested by centrifugation at 4000 rpm for 10 min and then resuspended in an infiltration working solution containing 5% sucrose and 0.02% Silwet L-77. The OD_600_ of the suspension was adjusted to 0.8, and the mixture was allowed to stand in the dark for 3 h.

*A. thaliana* was transformed using the floral dip method. Healthy plants were selected, and their primary inflorescences (1–5 cm long) were removed to encourage lateral branching. Before infiltration, opening flowers and developing siliques were excised from the plants at the full-bloom stage. Inflorescences were then immersed in the infiltration medium for 30 s, blotted dry, enclosed in black plastic bags, and incubated horizontally in the dark for 24 h. Following this procedure, the plants were transferred to a growth chamber under standard conditions until seed maturity. Mature seeds were surface-sterilized and sown on MS solid medium supplemented with 40 mg/L hygromycin (Hyg) for selection. Seedlings that showed resistance were further verified by PCR analysis to confirm the presence of the transgene.

### 2.8. Determination of Total Selenium Content

Total selenium content in plant samples was determined using hydride generation-atomic fluorescence spectrometry (HG-AFS). Exactly 0.1 g of sample was weighed into a digestion tube, then 10 mL HNO_3_ and 2 mL H_2_O_2_ were added. The mixture was vortexed to ensure complete suspension of sample particles adhering to the tube wall. Microwave-assisted digestion was performed using an intelligent microwave digester with the following program: 120 °C for 8 min, 150 °C for 5 min, and 180 °C for 20 min. After microwave digestion, the tubes were cooled to room temperature before adding 5 mL concentrated hydrochloric acid. The mixture was transferred to a graphite digester and heated at 180 °C until the solution became clear and colorless, with white fumes evolving. Upon cooling, the solution was transferred to a 10 mL centrifuge tube, mixed with 2.5 mL of 100 g/L potassium ferricyanide solution, and brought to volume with water. The homogenized mixture was filtered through a membrane filter prior to analysis. A series of selenium standard solutions (0, 5, 10, 20, 30 μg/L) was prepared from a stock standard (purchased from the National Institute of Metrology, Beijing, China) for calibration curve construction. During measurements, 5% hydrochloric acid was used as the carrier solution, and the reducing agent consisted of a mixture of 2% potassium borohydride and 5 g/L potassium hydroxide.

### 2.9. Determination of Selenium Speciation

Selenium speciation in plant samples was analyzed using liquid chromatography-atomic fluorescence spectrometry (LC-AFS). Exactly 0.2 g of freeze-dried *A. thaliana* samples was weighed, and streptomycin sulfate was added. The mixture then underwent ultrasound-assisted hydrolysis at 37 °C for 35 min. After hydrolysis, the products were centrifuged at 8000 rpm for 18 min, and the supernatant was collected and filtered through a membrane for subsequent analysis. Five selenium species standards, including selenite (SeO_3_^2−^), selenate (SeO_4_^2−^), selenocystine (SeCys_2_), selenomethionine (SeMet), and methylselenocysteine (MeSeCys) (all purchased from the National Institute of Metrology, China), were used for calibration curve construction. The operating parameters of LC-AFS were set as follows: the mobile phase consisted of a mixture of 40 mmol/L ammonium dihydrogen phosphate (NH_4_H_2_PO_4_) and 20 mmol/L potassium chloride (pH 6.0); the column temperature of PRPX100 chromatographic column was maintained at 25 °C; the injection volume was 100 μL; and the instrument parameters were set to a cathode current of 80 mA, negative high voltage of 400 V and a carrier gas flow rate 600 mL/min.

### 2.10. Statistical Analysis

Each treatment included three biological replicates, with gene relative expression levels computed via the 2^−ΔΔCT^ method [40]. All experimental data are presented as the mean ± standard deviation (SD) of three biological replicates. Statistical analyses were conducted using Excel 2023, while one-way analysis of variance (ANOVA) followed by Tukey’s post hoc test was performed in Origin 2024 to assess intergroup differences. Statistical significance was defined as *p* < 0.05.

## 3. Results

### 3.1. Identification and Chromosomal Localization of BpSULTR Gene Family

In this research, 9 sulfate transporters (BpSULTRs) were identified from the *B. papyrifera* genome. Analyses of their physicochemical properties (Table 1) show that the number of encoded amino acids ranges from 598 to 1486, with molecular weights between 65.26 and 166.94 kDa. Among them, *BpSULTR3;3* is the longest, while *BpSULTR3;2* is the shortest. The isoelectric points of BpSULTR members range from 6.23 to 9.13, indicating a high proportion of basic amino acids. With gravy values ranging from 0.037 to 0.533, all BpSULTRs are hydrophobic proteins. All members except *BpSULTR1;2* (instability index: 42.04) have instability indices between 32.74 and 39.91, indicating they are stable proteins. The subcellular localization prediction results of BpSULTR proteins showed that they were all located on the membrane. TMHMM predictions indicated that all family members possess 8–12 transmembrane domains, with the fewest in subfamilies I and II and the most in subfamily IV. According to SPOMA predictions (Figure 1a), the secondary structure of BpSULTRs mainly consists of α-helices (43.81–52.51%), extended strands (9.25–19.78%), and random coils (36.41–42.71%), with α-helices being the most abundant. As α-helices often serve as transmembrane regions, their hydrophobic amino acid side chains interact with the lipid bilayer, ensuring stable embedding of the proteins in the cell membrane to facilitate transmembrane transport and signal transduction. Tertiary structure models generated using SWISS-MODEL homology modeling (Figure 1a) further demonstrated that BpSULTRs contain abundant α-helices and random coils, forming a multi-layered spatial conformation. Chromosomal localization analysis using Tbtools software showed that the 9 *BpSULTR* genes are unevenly distributed across five chromosomes (Figure 1b). Chromosomes 1, 2, 3, and 13 each contain two genes, while *BpSULTR3;4* is the only gene on chromosome 7.

### 3.2. Classification and Phylogenetic Analysis of the BpSULTRs

To investigate the evolutionary characteristics of the BpSULTR family genes, we performed multiple sequence alignments between BpSULTRs from *B. papyrifera* and 14 sulfate transporters from *A. thaliana*. The e amino acid sequence alignment results (Appendix A) showed that all 9 BpSULTR proteins have typical sulfate transporter domains at both their *N*-terminus and *C*-terminus, including transmembrane domains (TMDs; Membrane-spanning domain) and STAS domains (Sulfate Transporter and Anti-sigma Antagonist domain). This confirms that they possess the canonical features of sulfate transporters. In contrast, Arabidopsis AtSULTR5;1 and AtSULTR5;2 lack these characteristic domains, which may explain their divergence from the other 12 *A. thaliana* SULTR members. Phylogenetic tree analysis (Figure 2) further revealed that a total of 23 SULTR proteins from *B. papyrifera* and *A. thaliana* together clustered into five evolutionary groups. Among these, the BpSULTR proteins of *B. papyrifera* were only distributed in the first four groups: Group I contained *BpSULTR1;1* and *BpSULTR1;2*; Group II included *BpSULTR2;1* and *BpSULTR2;2*; Group III comprised *BpSULTR3;1* to *BpSULTR3;4*; and Group IV contained only *BpSULTR4;1*. Notably, the majority of SULTR proteins from both *B. papyrifera* and *A. thaliana* were assigned to Group III, suggesting potential functional conservation of this group in sulfate transport. In addition, *B. papyrifera* contained only one BpSULTR member in Group IV and no members in Group V. In contrast, *A. thaliana* had two homologous proteins (AtSULTR5;1 and AtSULTR5;2) in Group V. This difference indicates a significant evolutionary divergence between the SULTR proteins of *B. papyrifera* and the AtSULTR5 subfamily members of *A. thaliana*.

### 3.3. Cis-Acting Element Analysis of BpSULTR Gene Promoters

A total of 144 cis-acting elements in eight categories were identified in the prediction performed from 2000 bp upstream sequences of *BpSULTRs* (Figure 3). The most common elements were light-responsive elements, with 50 elements found in 9 *BpSULTR* genes, suggesting that light may induce their expression. Additionally, 38 abscisic acid-responsive elements were found in the promoters of 9 *BpSULTR* genes, implying a role in adapting to abiotic stresses through abscisic acid. In addition, 30 methyl jasmonate-responsive elements were identified in seven *BpSULTR* gene promoters, indicating possible involvement in methyl jasmonate-related signaling pathways. Other elements, including eight salicylic acid-responsive, seven defense and stress-responsive, five gibberellin-responsive, five low-temperature-responsive, and one wound-responsive element, were unevenly distributed among the 9 *BpSULTR* gene promoters. Collectively, these findings suggest that *BpSULTR* genes may participate in relevant regulatory and physiological metabolic processes by responding to various hormonal and environmental signals.

### 3.4. Correlation Analysis Between the Expression Levels of BpSULTRs and Selenium Content

This study used RT-qPCR to examine the expression levels of 9 *BpSULTR* genes in *B. papyrifera* leaves treated with different concentrations of sodium selenate. As shown in Figure 4, a low concentration of 0.2 mmol/L Na_2_SeO_4_ led to a significant increase in the expression level of *BpSULTR4;1*, but not in the other *BpSULTR* genes. At an intermediate concentration of 0.4 mmol/L Na_2_SeO_4_, the expression levels of *BpSULTR2;1*, *BpSULTR3;1*, *BpSULTR3;2*, *BpSULTR3;4*, and *BpSULTR4;1* all significantly increased, while the remaining 4 *BpSULTR* genes (*BpSULTR1;1*, *BpSULTR1;2*, *BpSULTR2;2* and *BpSULTR3;3*) showed an upward trend. When the concentration of Na_2_SeO_4_ was increased to 0.8 mmol/L, the expression levels of *BpSULTR1;1*, *BpSULTR3;3*, and *BpSULTR3;4* were further increased, whereas the expression levels of *BpSULTR1;2*, *BpSULTR2;1*, *BpSULTR2;2*, *BpSULTR3;1*, and *BpSULTR3;2* were significantly reduced. These results suggest that *BpSULTRs* exhibit different response patterns to varying concentrations of Na_2_SeO_4_. *BpSULTR1;1*, *BpSULTR3;3* and *BpSULTR3;4* are more tolerant to high concentrations of Na_2_SeO_4_ and may be involved in the selenium tolerance response of *B. papyrifera*. In contrast, *BpSULTR1;2*, *BpSULTR2;1*, *BpSULTR2;2*, *BpSULTR3;1* and *BpSULTR3;2* are more sensitive to 0.4 mmol/L Na_2_SeO_4_ and may function as key selenium-enrichment genes in *B. papyrifera*, participating in its selenium metabolism process.

To further explore the relationship between *BpSULTR* expression and total Se content, we measured the total Se content in the leaves of *B. papyrifera* treated with different Na_2_SeO_4_ concentrations. The results indicated a continuous increase in total selenium content with higher concentrations, reaching values of 0.60, 52.78, 94.68, and 129.99 mg/kg DW, respectively. We then analyzed the correlation between *BpSULTR* gene expression levels and total Se content. The results (Figure 4) showed that among the 9 *BpSULTR* genes, *BpSULTR3;1* and *BpSULTR3;4* exhibited the highest correlations with total Se content, with *BpSULTR3;4* demonstrating a correlation of 0.630 and *BpSULTR3;1* showing a correlation of 0.480. These genes are likely involved in selenium accumulation in *B. papyrifera*. Therefore, we selected *BpSULTR3;1* and *BpSULTR3;4* for further functional validation.

### 3.5. Subcellular Localization Analysis

To analyze the subcellular localization of *BpSULTR3;1* and *BpSULTR3;4*, we conducted transient expression assays in *N. benthamiana*. Recombinant plasmids pICH86988-GFP, pICH86988-*BpSULTR3;1*-GFP, pICH86988-*BpSULTR3;4*-GFP, and the plasma membrane marker pmCFP were transformed into *Agrobacterium tumefaciens* and co-infiltrated into *N. benthamiana* leaves. Confocal laser scanning microscopy analysis revealed that the green fluorescence signals from *BpSULTR3;1*-GFP and *BpSULTR3;4*-GFP co-localized with the blue fluorescence of the plasma membrane marker, resulting in cyan-colored overlays (Figure 5). These results unambiguously demonstrate that both *BpSULTR3;1* and *BpSULTR3;4* are plasma membrane-localized proteins, corroborating the bioinformatics predictions.

### 3.6. Identification of Transgenic A. thaliana and BpSULTR3;1/BpSULTR3;4 Expression Analysis

The plant fusion expression vector pCYH05252-*BpSULTR3;1/3;4* (Figure 6a) was successfully constructed and then transformed into *Agrobacterium tumefaciens* strain GV3101. Generation of transgenic *A. thaliana* plants harboring the target gene using the floral dip method. Positive seedlings were screened and identified on MS solid medium supplemented with 40 mg/L hygromycin. As shown in Figure 6c, only a few positive seedlings were obtained from the harvested T_1_ generation of Arabidopsis. However, continuous screening and propagation of these positive seedlings up to the T_3_ generation yielded a large number of positive plants. PCR validation was further performed on the selected positive Arabidopsis plants, which confirmed that all of them could amplify the target gene band. In contrast, no distinct bands were detected in Arabidopsis transformed with the empty vector pCYH05252 (pCY) alone (Figure 6b). Analysis of the gene expression level in the identified T_3_ generation Arabidopsis plants revealed that (Figure 6d) the target genes could be detected in each of the three selected transgenic lines of *BpSULTR3;1* and *BpSULTR3;4*, whereas no gene expression was observed in Arabidopsis plants transformed with the empty vector. The results of gene expression level analysis of different Arabidopsis lines before and after treatment with 0.4 mmol/L Na_2_SeO_4_ showed that (Figure 6e) compared with the pre-treatment period, the expression levels of *BpSULTR3;1* and *BpSULTR3;4* increased significantly after selenate treatment. These results indicate that we successfully obtained transgenic Arabidopsis plants, and 0.4 mmol/L Na_2_SeO_4_ treatment can significantly promote the expression of the target genes.

### 3.7. Analysis of Total Se Content and Selenium Speciation of Transgenic A. thaliana

Total Se content and selenium speciation were determined in T_3_ transgenic *A. thaliana* plants after selenium treatment. The results showed that (Figure 7a) the total selenium content in *BpSULTR3;1* transgenic *A. thaliana* increased significantly compared to the control group, with an average total Se content of 185.25 mg/kg DW, 2.31 times that of the control group (80.06 mg/kg DW). In contrast, there was no significant difference in total selenium content between *BpSULTR3;4* transgenic *A. thaliana* and the control group. We identified five selenium speciations, as illustrated in Figure 7b, with the results presented in Figure 7c,d. It is evident that five distinct selenium speciations were detected in both the control group and the *BpSULTR3;1* transgenic *A. thaliana*, while MeSeCys was not observed in *BpSULTR3;4*, likely due to its concentration being below the instrument’s detection limit. Notably, in both the control group and the transgenic Arabidopsis, SeMet emerged as the most abundant organic selenium form, with concentrations of 9.08 mg/kg DW, 16.9 mg/kg DW, and 7.64 mg/kg DW, respectively. In contrast, SeO_4_^2−^ was the predominant inorganic selenium form, with concentrations of 68.05 mg/kg DW, 169.77 mg/kg DW, and 76.53 mg/kg DW, respectively. Comparative analysis revealed that the SeMet content in *BpSULTR3;1* transgenic *A. thaliana* was 1.86 times greater than that of the control group, while the SeO_4_^2−^ content was 2.49 times higher. The findings indicate that the overexpression of *BpSULTR3;1* significantly enhances the transport and uptake of selenate in *A. thaliana*, while *BpSULTR3;4* exhibits no significant effect.

### 3.8. Expression Pattern Analysis of Selenium Metabolism-Related Genes in Transgenic A. thaliana

The uptake and translocation of selenium by plants is an extremely complex process that involves multiple enzymes. This study analyzed the expression levels of key enzyme genes involved in selenium metabolism, with the results presented in Figure 8. Overexpression of *BpSULTR3;1* in *A. thaliana* significantly increased the expression of Selenocysteine methyltransferases (SMT), an enzyme that efficiently converts SeCys to form MeSeCys, leading to increased MeSeCys accumulation in the transgenic plants. Concomitantly, the expression levels of three adenylyl-sulfate reductase (APR) genes were markedly elevated. This increase is attributed to *BpSULTR3;1*-mediated enhancement of selenate uptake and translocation into plant cells, which induces the substantial expression of downstream APRs. These enzymes reduce accumulated selenate to selenite, fueling plant selenium metabolic pathways. Furthermore, further analysis of the data presented in Figure 8 revealed that in transgenic *A. thaliana* plants overexpressing *BpSULTR3;1*, among the four S-adenosylmethionine synthetase (SAM) genes examined, only the expression level of *AtSAM1* exhibited a statistically significant upregulation compared with the control group. In contrast, the expression levels of the remaining three SAM genes (e.g., *AtSAM2*/*AtSAM3*/*AtSAM4*) showed no statistically significant differences relative to the control group. Meanwhile, the expression level of the sole methionine S-methyltransferase (MMT) gene (*AtMMT*) in the plants also displayed no statistically significant increase. In contrast, the analysis of the expression profiles of related genes following the overexpression of *BpSULTR3;4* revealed that, among the 15 selected enzyme-encoding genes, only three homocysteine S-methyltransferases (HMT) genes were significantly upregulated compared to the control, while *AtSMT* and *AtAPR*-encoding genes were significantly downregulated, and the remaining 6 genes showed no significant differences in expression. Furthermore, a comparison of the expression levels of orthologous enzyme-encoding genes in the selenium metabolic pathway between *A. thaliana* overexpressing *BpSULTR3;1* and *BpSULTR3;4* indicated opposing regulatory trends, suggesting that these two transporters may serve distinct functional roles.

## 4. Discussion

In our study, we identified 9 SULTRs from the whole genome of *B. papyrifera*. Bioinformatics analyses showed that all 9 BpSULTR proteins have both TMDs and STAS domains, indicating a high degree of conservation. We performed a phylogenetic analysis of these BpSULTR proteins along with 14 SULTR proteins from *A. thaliana*, which classified the 9 BpSULTRs into four groups. This is consistent with the evolutionary classification of SULTRs in other plant species [41,42]. Notably, Group III is the largest group, containing four BpSULTRs, suggesting that it may perform more important functions. However, some studies have identified a fifth group of SULTR proteins in plants, including *A. thaliana* AtSULTR5;1 (At1g80310), AtSULTR5;2 (At2g25680), and *Oryza sativa* OsSULTR5;1 and OsSULTR5;2 [43,44]. Although these proteins differ significantly from the Group I to Group IV, for example, they lack the STAS domain, they are still distinctly related and cannot be entirely separated from the other groups [29]. To date, the activity of SULTR proteins has only been experimentally confirmed in Groups I and II. Due to the high homology among family members, it is reasonable to consider all these proteins as integral members of the SULTR family genes.

Analysis of the cis-acting elements within the promoter regions of 9 *BpSULTR* genes showed that they contain multiple light-responsive elements and response elements for various plant hormones, including abscisic acid, salicylic acid, and methyl jasmonate. Moreover, these genes may respond to abiotic stresses such as low temperature, wounding, pathogen defense, and mechanical pressure, consistent with previous studies [33,45]. These results collectively suggest that *BpSULTR* genes may play important roles in resisting salt stress. This is supported by the expression patterns of the 9 *BpSULTR* genes under different concentrations of sodium selenate treatment: five genes exhibited significantly increased expression levels at 0.2 mmol/L, 0.4 mmol/L, and 0.8 mmol/L compared to the control group, consistent with the response characteristics of SULTR genes in tea plants [46]. However, distinct *BpSULTR* members showed differential sensitivity to selenate concentrations, which may be attributed to variations in their affinity for selenate [26,30,36]. Similarly, studies on *ZmSULTR* genes have demonstrated that their expression levels are significantly upregulated under stress conditions [47]. In *A. thaliana*, the expression of *AtSULTR3;1* is markedly induced under drought and salt stress [33]. Collectively, these findings indicate that plant SULTRs are involved in the response to salt stress.

The localization of a protein often determines its functional role. In our study, we analyzed the subcellular localization of two SULTR proteins, *BpSULTR3;1* and *BpSULTR3;4*, and found that they both function at the plant plasma membrane, similar to TaSULTR1s in tobacco [48]. Interestingly, the localization of SULTR proteins varies significantly across different species. For example, in *A. thaliana*, Group I SULTR1s are localized to the plasma membrane, Group III to the chloroplast membrane, and Group IV to the vacuolar membrane [36,49]. Similarly, other studies have shown that GmSULTR1;2a from soybean is localized to the plasma membrane [50]; CsSULTR3.1 from tea plants is primarily distributed in the cytoplasm, and CsSULTR3.5 is localized to the plasma membrane [46]. JrSULTR1.2b and JrSULTR3.1a from walnut are also localized to the plasma membrane [51]. These diverse localization patterns suggest that SULTRs have functional divergence: plasma membrane-localized SULTRs, such as CsSULTR3.5, *BpSULTR3;1*, and *BpSULTR3;4*, likely play a direct role in sulfate uptake from the external environment, whereas cytoplasm-localized SULTRs, such as CsSULTR3.1, may be involved in regulating intracellular sulfur. Furthermore, the co-localization of SULTR3;5 and SULTR2;1 in Arabidopsis reveals a synergistic interaction among SULTR family members, where SULTR3;5 enhances sulfate loading efficiency in the xylem by regulating SULTR2;1 stability [32].

Selenate is a primary form of selenium in nature, actively absorbed by plants through sulfate transporters. In this study, we successfully generated *A. thaliana* plants overexpressing *BpSULTR3;1* and *BpSULTR3;4*. These transgenic plants were treated with 0.4 mmol/L selenate and then their total selenium content and selenium speciation were determined. The total selenium content in *BpSULTR3;1*-overexpressing Arabidopsis was 2.31 times s higher than that in the pCY control and 2.34 times higher than *BpSULTR3;4*-overexpressing *A. thaliana*. The content of organic selenium species (SeCys_2_, MeSeCys, and SeMet) in *BpSULTR3;1*-overexpressing lines was 1.92 times higher than the pCY control and 2.46 times higher than *BpSULTR3;4*-overexpressing *A. thaliana*, with statistically significant differences. Notably, *BPSULTR3;4* overexpression plants absorbed selenium at a significantly lower rate than *BpSULTR3;1*, which may be attributed to the tissue-specific expression of SULTRs in plants. The study of SULTRs in Arabidopsis has confirmed this point, with *AtSULTR1;1* and *AtSULTR1;2* involved in selenate absorption in roots [30], and *AtSULTR4;1* and *AtSULTR4;2* involved in transporting selenate out of the vacuole [34]. Similarly, in other plant species, *LaSULTR2;1*, *LaSULTR2;2* and *LaSULTR3;5* contribute to the long-distance transport of selenate− from root to shoot [52,53]. However, due to the similar physicochemical properties and metabolic pathways shared by sulfur and selenium, SULTRs’ absorption and transport of selenate are also influenced by sulfate. Many studies have shown that selenate enters plants by competing with sulfate for plasma membrane-localized SULTRs [54]. Chen et al. also found that overexpression *OsSULTR1;1* under low-sulfur conditions promotes selenium uptake in such environments [55]. High concentrations of sulfate can inhibit plant absorption of selenate to some extent [25]. In selenium hyperaccumulators, SULTRs tend to prefer selenium absorption, and the expression levels of SULTRs differ between hyperaccumulators and non-accumulators [56,57]. Overall, the selectivity of SULTRs for selenate and sulfate is affected by external sulfate concentration, plant species, and transporter type [58].

SULTRs are crucial for selenate uptake into plant cells. Overexpressing *BpSULTR3;1* substantially alters the expression of numerous genes involved in selenium metabolism, whereas *BpSULTR3;4* has a relatively milder impact. The increased expression of *BpSULTR3;1* is likely due to its higher affinity for selenate and more efficient transport across the plasma membrane [58]. This enhanced transport increases the intracellular selenate concentration, triggering the up-regulation of downstream genes involved in selenium metabolism. In contrast, *BpSULTR3;4* may have different transport properties, resulting in less significant effects on Se metabolic pathway [59]. Adenylyl-sulfate reductase (APR) is a key enzyme in Se metabolism, catalyzing the reduction of selenate to selenite [60,61]. Overexpressing *BpSULTR3;1* significantly up-regulates the expression of APR genes (*AtAPR1*, *AtAPR2*, *AtAPR3*), including that *BpSULTR3;1* promotes selenate influx and stimulates its reduction [62]. As a result, *BpSULTR3;1* increases the demand for selenate reduction, accelerating its conversion to selenite, which is necessary for further Se metabolism, such as organic Se synthesis. In contrast, *BpSULTR3;4* has a weaker effect on APR gene expression, suggesting a lesser ability to drive selenate reduction. Selenocysteine methyltransferase (SMT) catalyzes the conversion of selenocysteine to less toxic and more stable organic Se compounds like MeSeCys [63]. In the *BpSULTR3;1* overexpression group, the up-regulation of *AtSMT1*, *AtSMT2* and *AtSMT3* genes implies that *BpSULTR3;1* promotes not only selenate uptake and reduction but also facilitates organic Se synthesis. This coordinated regulation ensures that the increased selenate entering the cell is efficiently converted into organic forms, reducing the potential toxicity of free selenate [64]. The limited effect of *BpSULTR3;4* on SMT genes further emphasizes the functional differences between these two SULTRs in Se metabolism. S-adenosylmethionine synthetase (SAM) is involved in both sulfur and selenium metabolism, as these two elements share similar metabolic pathways. The significant upregulation in *AtSAM1* expression in *BpSULTR3;1*-overexpressing plants suggests that *BpSULTR3;1* may modulate the metabolic balance between sulfur and selenium. Additionally, *BpSULTR3;1* and *BpSULTR3;4* exert distinct effects on other selenium metabolism—related genes, such as HMT, Nifs, and MMT. This is most likely due to differences in their protein structures, transport activities, and substrate affinities. *BpSULTR3;1* appears to be a more potent regulator of Se metabolism, exerting a broader and more pronounced influence on gene expression across multiple Se metabolic processes, including uptake, reduction, and organic synthesis. In contrast, *BpSULTR3;4* has a more limited or variable role, potentially due to lower transport efficiency for selenate, distinct substrate preferences, or a requirement for specific regulatory factors or protein—protein interactions to exert its full functional capacity.

## 5. Conclusions

We identified 9 sulfate transporters in the complete genome of *B. papyrifera*. Based on the correlation between gene expression levels following treatment with varying concentrations of selenate and the total selenium content, *BpSULTR3;1* and *BpSULTR3;4* were selected for further study. Functional verification experiments demonstrated that the overexpression of *BpSULTR3;1* significantly enhanced selenate uptake by the plants and influenced several downstream processes, including selenate reduction and organic selenium synthesis, thus playing a crucial regulatory role in the selenium metabolic network. In contrast, variations in protein structure, transport activity, and substrate affinity likely explain why *BpSULTR3;4* may not participate in selenate transport or may have only a minimal effect; its precise function requires further investigation.

## Figures and Tables

**Figure 1 plants-14-02995-f001:**
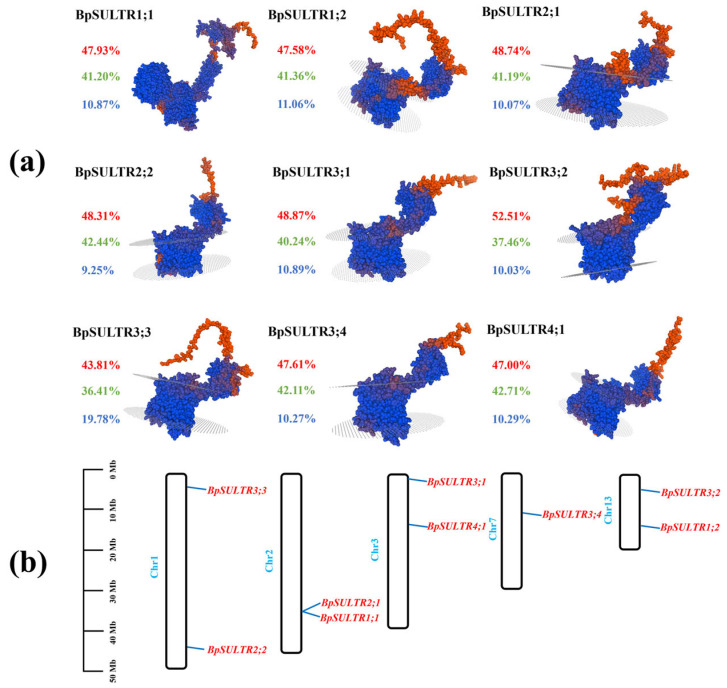
Protein structure prediction and chromosomal localization of BpSULTRs in *B. papyrifera*: (**a)** Protein structure prediction of BpSULTRs. The colored percentage values represent the content of three distinct secondary structures: red denotes *α*-helix, green denotes random coil, and blue denotes extended strand. (**b**) Distribution of BpSULTRs on chromosomes.

**Figure 2 plants-14-02995-f002:**
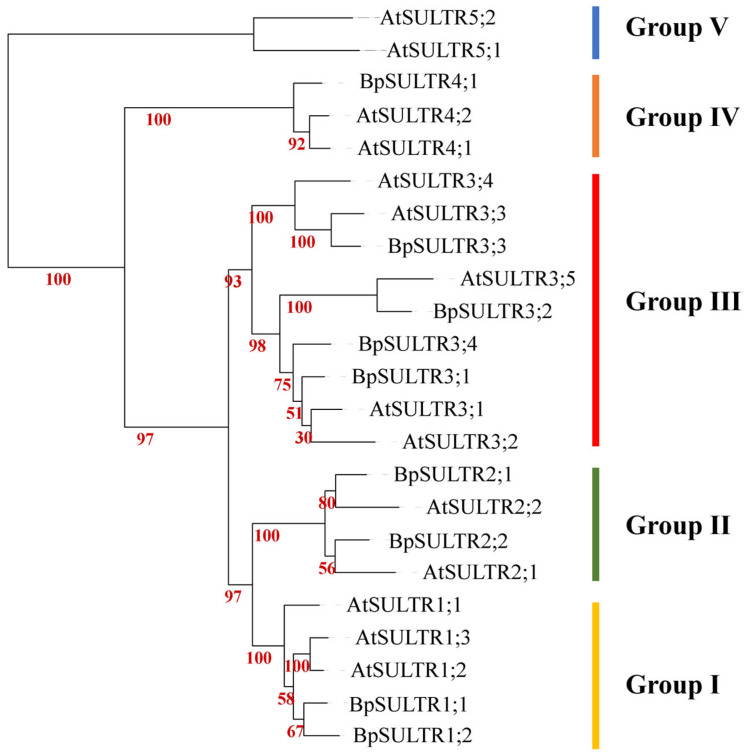
Phylogenetic tree of SULTR family genes. Coarse lines of different colors represent different groups. Yellow: Group I, Green: Group II, Red: Group III, Orange: Group IV, Blue: Group V.

**Figure 3 plants-14-02995-f003:**
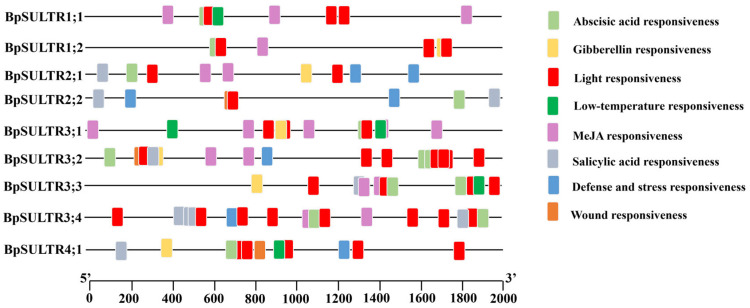
Cis-acting element analysis of *BpSULTR* promoter. Different colored squares represent different response elements.

**Figure 4 plants-14-02995-f004:**
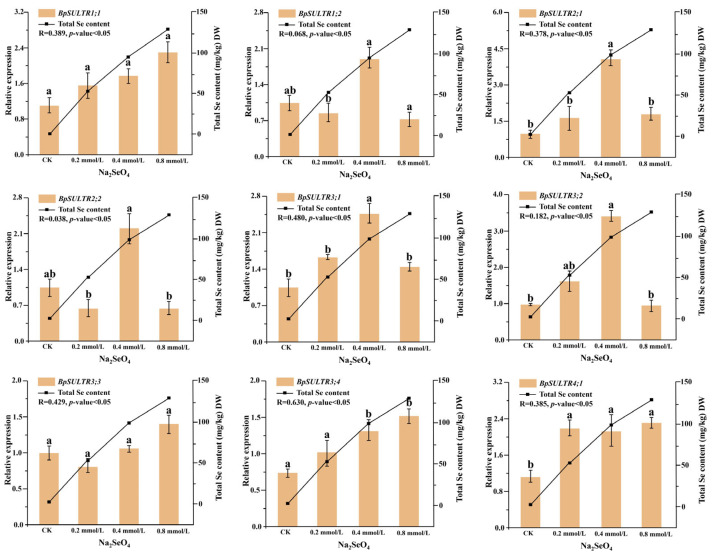
The expression levels of *BpSULTR* genes in leaves of *B. papyrifera* under different Na_2_SeO_4_ treatments. Relative expression levels on the left correspond to the left *Y*-axis of all subplots, represented as orange bars. The total selenium content on the right corresponds to the right *Y*-axis of all subplots and is depicted as a line graph. R-values indicate the correlation between *BpSULTR* genes and total Se content, with a *p*-value < 0.05 indicating high significance. The axis scales of each subplot are adapted based on the expression range of the corresponding gene, and the specific values are shown in the text. Lowercase letters “a” and “b” denote a significant difference between treatments (*p* < 0.05).

**Figure 5 plants-14-02995-f005:**
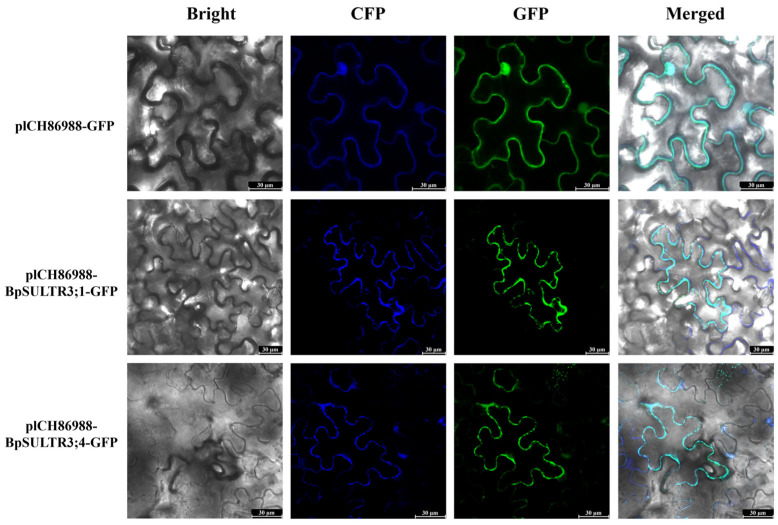
Subcellular localization of *BpSULTR3;1* and *BpSULTR3;4*. pICH86988-GFP: empty vector, pICH86988-BpSULTR3;1/BpSULTR3;4: target fusion protein, GFP: eGFP fluorescence, CFP: plasma membrane marker, Bright: bright-field images, Merged: overlay images.

**Figure 6 plants-14-02995-f006:**
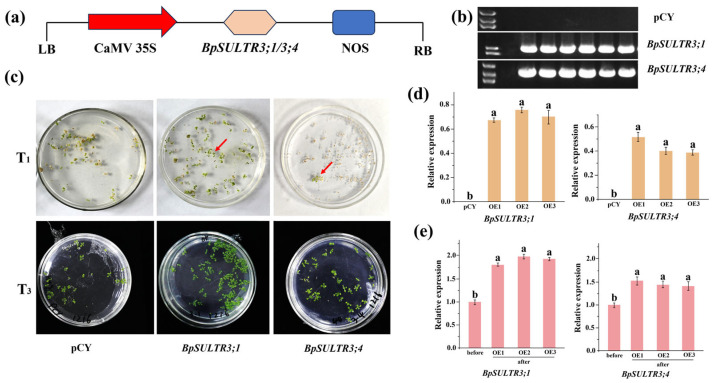
Screening and identification of transgenic *A. thaliana*: (**a**) Construction of pCYH05252-*BpSULTR3;1/3;4* vector. (**b**) PCR analysis in transgenic *A. thaliana* overexpressing *BpSULTR3;1/3;4*. (**c**) Resistance screening of transgenic *A. thaliana*. (**d**) Expression level analysis of three homozygous lines of T_3_ transgenic *A. thaliana*. (**e**) Expression level of *BpSULTR3;1/3;4* before and after selenium treatment. The red arrow indicates the screened transgenic lines. Lowercase letters “a” and “b” denote a significant difference between treatments (*p* < 0.05).

**Figure 7 plants-14-02995-f007:**
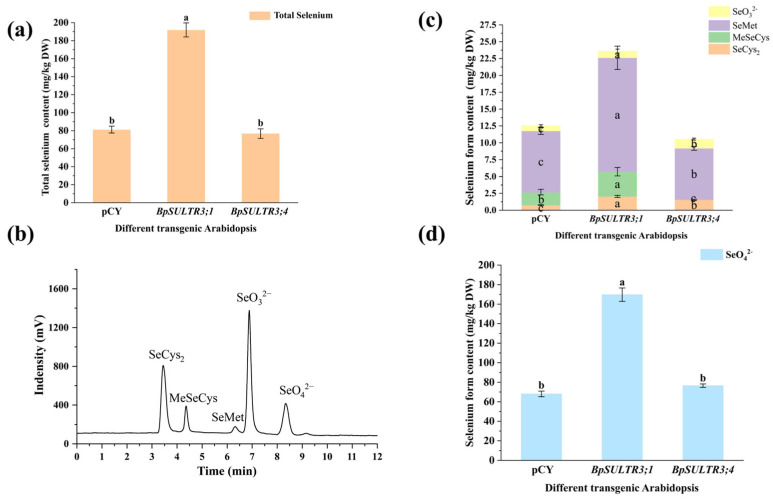
Total selenium content and selenium speciation in *BpSULTR3;1/3;4* transgenic A. thaliana: (**a**) Total selenium content in transgenic *A. thaliana*. (**b**) The standard HPLC curve for SeCys_2_, MeSeCys, SeMet, SeO_3_^2−^ and SeO_4_^2−^. (**c**,**d**) Contents of various selenium speciation in transgenic *A. thaliana*. Lowercase letters “a” “b” ”c” denote a significant difference between treatments (*p* < 0.05).

**Figure 8 plants-14-02995-f008:**
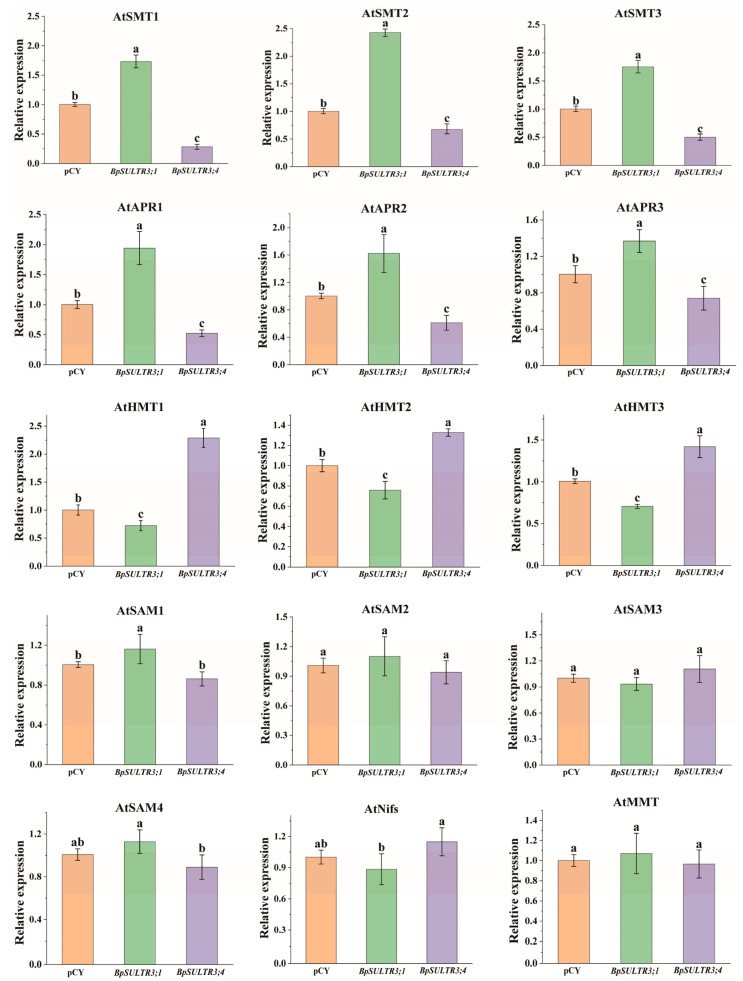
Relative expression levels of selenium-related genes in transgenic *A. thaliana*, including Selenocysteine methyltransferase (SMT), adenylyl-sulfate reductase (APR), Homocysteine S-methyltransferases (HMT), S-adenosylmethionine synthetase (SAM), Cysteine desulfurases (Nifs) and Methionine S methyltransferase (MMT). The axis scales of each subplot are adapted based on the expression range of the corresponding gene, and the specific values are shown in the text. Lowercase letters “a” “b” ”c” denote a significant difference between treatments (*p* < 0.05).

**Table 1 plants-14-02995-t001:** Physicochemical properties of BpSULTR proteins in *B. papyrifera*.

Gene Name	ID	Amino Acids	Molecular Weight (kDa)	pI	Instability Index	Aliphatic Index	Gravy	Subcellular Location
*BpSULTR1;1*	Bp01G0496.1	699	77.12	8.69	38.44	105.08	0.249	Membrane
*BpSULTR1;2*	Bp01G7368.1	660	73.05	9.06	42.04	109.32	0.347	Membrane
*BpSULTR2;1*	Bp02G6891.1	675	72.75	9.00	33.66	114.41	0.463	Membrane
*BpSULTR2;2*	Bp02G6906.1	681	73.66	8.75	32.74	108.87	0.400	Membrane
*BpSULTR3;1*	Bp03G0171.1	661	72.77	8.37	29.41	105.61	0.412	Membrane
*BpSULTR3;2*	Bp03G2059.1	598	65.26	8.81	36.12	111.37	0.533	Membrane
*BpSULTR3;3*	Bp07G1646.1	1486	166.94	6.23	33.86	95.37	0.037	Membrane
*BpSULTR3;4*	Bp13G0791.1	691	75.44	9.13	38.01	108.18	0.346	Membrane
*BpSULTR4;1*	Bp13G2065.1	700	76.45	8.29	39.91	111.84	0.357	Membrane

## Data Availability

All data generated or analyzed during this study are included in this article and its Appendix A.

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
