# Peer review of "Identification of Sulfate Transporter Genes in Broussonetia papyrifera and Analysis of Their Functions in Regulating Selenium Metabolism"

_plants, 2025, doi:10.3390/plants14192995_

Round 1

Reviewer 1 Report

Comments and Suggestions for Authors

Using standard bioinformatic methods and equipment, the authors have identified nine potential sulfate transporter genes in a deciduous shrub and looked for promising effects  on the selenium content in transgenic Arabidopsis thaliana.  Based on the effects they observe, they speculate that one of the family members, BpSULTR3:1, plays a crucial role in selenium metabolism.  While, undoubtedly, substantial technology has been applied in these studies and I have no serious reservations about the suggestions that the authors make, I feel that biologically significant conclusions cannot be drawn, both because many of the changes or effects are fairly modest and because there is no evidence that Arabidopsis is an appropriate model for the shrub they are studying.  I feel their ideas actually should be tested in their shrub or that, as proposed, the shrub be improved genetically, using the gene they have identified.  It seems strange that only one family member has any effect and the possibility of indirect effects have not been taken into consideration at all.  Seems to me, their concluding statement "precise function requires further investigation" sums up the paper well.  Accordingly, I feel this study may be too preliminary for publication in plants and, perhaps, more suitable for a specialized agricultural journal.

Author Response

Using standard bioinformatic methods and equipment, the authors have identified nine potential sulfate transporter genes in a deciduous shrub and looked for promising effects on the selenium content in transgenic Arabidopsis thaliana. Based on the effects they observe, they speculate that one of the family members, BpSULTR3;1, plays a crucial role in selenium metabolism.  While, undoubtedly, substantial technology has been applied in these studies and I have no serious reservations about the suggestions that the authors make, I feel that biologically significant conclusions cannot be drawn, both because many of the changes or effects are fairly modest and because there is no evidence that Arabidopsis is an appropriate model for the shrub they are studying. I feel their ideas actually should be tested in their shrub or that, as proposed, the shrub be improved genetically, using the gene they have identified. It seems strange that only one family member has any effect and the possibility of indirect effects have not been taken into consideration at all. Seems to me, their concluding statement "precise function requires further investigation" sums up the paper well. Accordingly, I feel this study may be too preliminary for publication in plants and, perhaps, more suitable for a specialized agricultural journal.

Response: At present, a stable and efficient genetic transformation system for B. papyrifera has not yet been established. Therefore, our transgenic function verification experiments have not been carried out on B. papyrifera. Arabidopsis thaliana is the main model plant used for genetic transformation verification. The evolutionary analysis and amino acid sequence alignment results showed that the functional domains of sulfate transporters between Arabidopsis and B. papyrifera are highly conserved, with a sequence similarity of over 70%, and Arabidopsis' such transporters can participate in plant selenium absorption via sulfate transport. So, in our research, transgenic experiments on Arabidopsis thaliana were conducted to indirectly demonstrate the function of BpSULTR genes in regulating selenium metabolism. Based on the quantitative data and their correlation with selenium, and the results of transgenic verification, we have drawn the research conclusion that BpSULTR3:1 plays a regulatory role in the selenium metabolism of B. papyrifera.

Reviewer 2 Report

Comments and Suggestions for Authors

In relation to the paper "Identification of sulfate transporter genes in Broussonetia papyrifera and analysis of their functions in regulating selenium  metabolism", In this article, the authors evaluate selenium/sulfur transporters in the species of interest through different approaches, including bioinformatics, evaluation of expression of hygiens of interest in the presence of different concentrations of Selenia, intracellular localization of two selected selenium transporter genes, as well as the functionality of these in transformed Arabidopsis plants, In the presence of selenium.

The article is an easy-to-read well-written article, with up-to-date references, and a proper introduction, which presents both the state of the art, in terms of sulfur/selenium transporters in various species, as well as the usefulness of this compound in a framework of human health and plant metabolism.

The materials and methods section are properly written with the necessary information for their reproducibility. It is only suggested to argue the reason for the use of the ubiquitin gene as a reference gene for the analysis of relative expression, in relation to the experimental differential of the Ct value of the reference gene and the various genes analyzed, since the validity of using it as a normalizer or reference gene depends on it.

In the results section, these are presented and analyzed correctly, however, for this reviewer, the presentation of the graphs of figures 4, 6 and 8 could be improved, including an axis on the same scale for relative expression, since being a multigraphic panel one of its functions is visual comparison so including different scales, It can lead to a misinterpretation of the information obtained at a glance, although it is important to mention that in the development of the text in the corresponding results some of the values that are seen in the graphs are clearly mentioned. This suggestion is made based on a better way of presenting the results, however, it is not necessary in a decisive way to do it, since the graphs are well planned.

There are some finger errors like on line 459 in the word leading.

For this reviewer, it is important to change the wording of the phylogenetic analysis paragraph from lines 316 to 322, since the text suggests that the proteins of the species of interest are classified into five groups. However, the study reports five groups, including the proteins of the species of interest and the proteins of the Arabidopsis, noting that the proteins of the species of interest are not present in group five, therefore, the the text is incorrect, since the proteins of the species of interest only formed or, are present, in four groups.

In relation to the discussion, this is well stated and compared against the literature and shows sufficient data with which the observations and conclusions of the study are supported.

It is suggested to include in the supplementary file with the primers, the Tm used for each gene, as useful information to replicate the experiments.

For this reviewer, this article may be accepted with minor, optional changes, and a review of typographical errors.

Author Response

  1. In relation to the paper "Identification of sulfate transporter genes in Broussonetia papyrifera and analysis of their functions in regulating selenium metabolism", In this article, the authors evaluate selenium/sulfur transporters in the species of interest through different approaches, including bioinformatics, evaluation of expression of hygiens of interest in the presence of different concentrations of Selenia, intracellular localization of two selected selenium transporter genes, as well as the functionality of these in transformed Arabidopsis plants, In the presence of selenium.

The article is an easy-to-read well-written article, with up-to-date references, and a proper introduction, which presents both the state of the art, in terms of sulfur/selenium transporters in various species, as well as the usefulness of this compound in a framework of human health and plant metabolism.

Response: Thank you for your comments on our manuscript. Your positive feedback and detailed recognition of our work have greatly encouraged our team.

  1. 2. The materials and methods section are properly written with the necessary information for their reproducibility. It is only suggested to argue the reason for the use of the ubiquitin gene as a reference gene for the analysis of relative expression, in relation to the experimental differential of the Ct value of the reference gene and the various genes analyzed, since the validity of using it as a normalizer or reference gene depends on it.

Response: Current research indicates that the BpUbiquitin gene is suitable to be selected as a reference gene for quantitative PCR experiments. For example, Chen et al. (2024) investigated the gene expression of the BpHMT2 gene in Broussonetia papyrifera under selenium treatment, and Zhu et al. (2025) conducted research on the involvement of MYB transcription factors in the selenium stress response of Broussonetia papyrifera. Their studies confirmed that the selected ubiquitin gene has stable expression characteristics and is a reliable reference gene. Subsequently, we verified its expression stability through preliminary experiments: there was no significant difference in the Ct values of the ubiquitin gene among all treatment groups in this study. Therefore, this study selected the ubiquitin gene as the reference gene for relative expression analysis.

  1. 3. In the results section, these are presented and analyzed correctly, however, for this reviewer, the presentation of the graphs of figures 4, 6 and 8 could be improved, including an axis on the same scale for relative expression, since being a multigraphic panel one of its functions is visual comparison so including different scales, It can lead to a misinterpretation of the information obtained at a glance, although it is important to mention that in the development of the text in the corresponding results some of the values that are seen in the graphs are clearly mentioned. This suggestion is made based on a better way of presenting the results, however, it is not necessary in a decisive way to do it, since the graphs are well planned.

Response: Thank you for your suggestions. We understand your intention to enhance the intuitive comparability of multi-subplot combinations and avoid misinterpretation of information by "unifying the scales for relative expression levels". We have checked the figure data, In the multi-subplot combinations of Figures 4, 6, and 8, there are significant differences in the ranges of relative expression levels of the genes to be analyzed among different subplots (e.g., the fluctuation range of gene expression levels in some subplots is 0.7–1.5, while that in other subplots is 0.8–4.5). If a unified scale were adopted, the trend of data changes in subplots with a narrow expression range would be excessively compressed, which would instead obscure key data details. So, we have kept the original display of the figures.

  1. There are some finger errors like on line 459 in the word leading.

Response: Thank you very much for your careful review and for pointing out the typo (finger error) in the manuscript, we have checked line 459 and corrected the typo in question. Additionally, to ensure there are no other similar errors, we have conducted a comprehensive, line-by-line proofreading of the entire manuscript—focusing on wording, spelling, and punctuation—to eliminate any potential inaccuracies that may have been overlooked. Corrections have been marked in red accordingly.

  1. For this reviewer, it is important to change the wording of the phylogenetic analysis paragraph from lines 316 to 322, since the text suggests that the proteins of the species of interest are classified into five groups. However, the study reports five groups, including the proteins of the species of interest and the proteins of the Arabidopsis, noting that the proteins of the species of interest are not present in group five, therefore, the text is incorrect, since the proteins of the species of interest only formed or, are present, in four groups.

Response: Thank the reviewer' s suggestions, we have rephrased this part of the content, with the revisions as follows:

Phylogenetic tree analysis (Figure 2) further revealed that a total of 23 SULTR proteins from B. papyrifera and A. thaliana together clustered into five evolutionary groups. Among these, the BpSULTR proteins of B. papyrifera were only distributed in the first four groups: Group I contained BpSULTR1;1 and BpSULTR1;2; Group II included BpSULTR2;1 and BpSULTR2;2; Group III comprised BpSULTR3;1 to BpSULTR3;4; and Group IV contained only BpSULTR4;1. Notably, the majority of SULTR proteins from both B. papyrifera and A. thaliana were assigned to Group III, suggesting potential functional conservation of this group in sulfate transport. In addition, B. papyrifera contained only one BpSULTR member in Group IV and no members in Group V. In contrast, A. thaliana had two homologous proteins (AtSULTR5;1 and AtSULTR5;2) in Group V. This difference indicates a significant evolutionary divergence between the SULTR proteins of B. papyrifera and the AtSULTR5 subfamily members of A. thaliana.

  1. In relation to the discussion, this is well stated and compared against the literature and shows sufficient data with which the observations and conclusions of the study are supported.

Response: Thank you for the reviewer's comments.

  1. It is suggested to include in the supplementary file with the primers, the Tm used for each gene, as useful information to replicate the experiments.

Response: Thank you for your suggestion. We added the Tm values corresponding to each gene in the newly submitted supplementary document.

  1. For this reviewer, this article may be accepted with minor, optional changes, and a review of typographical errors.

Response: Thank the reviewer's comments, we have also carefully revisited the entire manuscript again.

Reviewer 3 Report

Comments and Suggestions for Authors

The manuscript is devoted to the studies of the role of sulfate transporters in selenium absorption by Broussonetia papyrifera. Interesting and novel results that may have not only fundamental but also practical significance were obtained. The following comments have arisen when analyzing the manuscript:

  1. It is necessary to provide a description of the composition of the nutrient substrate on which Broussonetia papyrifera and Nicotiana benthamiana plants were grown.
  2. It is unclear how the concentrations of 0.2, 0.4, 0.8 mmol/L Na2SeO4 were selected. Were they toxic? Did they affect plant growth?
  3. It is unclear why different designs of Se treatments were used for Broussonetia papyrifera and Arabidopsis thaliana (lines 117, 133).
  4. A shortcoming of the manuscript is the lack of the data on the effects of Se treatments on plants, at least regarding the simplest morphometric or physiological parameters. Without such data, it is difficult to assess the effects of the applied Se treatments on plants. Therefore, these data should be incorporated.
  5. It is necessary to check the spelling of the Latin names of the species in italics throughout the text (e.g., line 207).
  6. The authors indicate that "Figure 8 also shows that the expression of three S-adenosylmethionine synthetase (SAM) genes and one methionine S methyltransferase (MMT) gene was moderately increased" (lines 464-466). However, this increase is insignificant in most cases except for SAM1 according to the data shown in Figure 8. It is important to correct the text (in Results, Discussion, etc.) according to the data of the statistical analysis.

Author Response

  1. It is necessary to provide a description of the composition of the nutrient substrate on which Broussonetia papyrifera and Nicotiana benthamiana plants were grown.

Response: Thank you for your valuable comment, we have added the composition of the nutrient substrate for both plant species and supplemented the details in the "Materials and Methods" section:

For Broussonetia papyrifera: These seedlings were planted in circular pots with a height of 14 cm and a top diameter of 12 cm. The nutrient substrate was a mixture of peat, red soil, vermiculite, and perlite at a volume ratio of 6:1:1:1. All seedlings were uniformly cultivated in greenhouse of Yangtze University, Hubei (30° 37' N,112° 07' E). The growth conditions were set as follows: ambi-ent temperature of 25 °C, relative humidity of 65%, and a photoperiod of 12 h light/12 h dark. Meanwhile, regular water and fertilizer management were conducted throughout the cultivation period.

For Nicotiana benthamiana: Nicotiana benthamiana was used for subcellular localization assays. Prior to use, its seeds were stored in a refrigerator at 4°C. The seeds were then sown in a nutrient substrate consisting of peat, vermiculite, and perlite at a volume ratio of 7:2:1. Then seedlings were transplanted at the 3-4 true-leaf stage.

  1. It is unclear how the concentrations of 0.2, 0.4, 0.8 mmol/L Na2SeO4 were selected. Were they toxic? Did they affect plant growth?

Response: Thank you for the reviewer's comments. The selection of Na2SeO4 treatment concentrations (0.2, 0.4, and 0.8 mmol/L) in this study was primarily based on two lines of evidence: On the one hand, prior to the initiation of this study, we had systematically investigated the effects of different selenium sources and treatment concentrations on the growth and development of Broussonetia papyrifera. Relevant findings have been published in the journal Tree Physiology (Chen et al., 2022), which provided a foundation for the preliminary screening of concentration ranges. On the other hand, before the formal experiment, we referenced research results of sodium selenate application in other plant species, set up multiple concentration gradients, and conducted preliminary experiments. Finally, the above three gradients were determined.

Numerous existing studies have confirmed that sodium selenate at certain concentrations can promote plant growth and development and improve their nutritional quality (Arı et al., 2022; Li et al., 2022; Poblaciones et al.,2024). In this study, the three concentrations of sodium selenate exerted different effects on Broussonetia papyrifera: 0.2 mmol/L significantly promoted the growth of B. papyrifera; 0.4 mmol/L showed no significant promoting or inhibitory effect; when the concentration increased to 0.8 mmol/L, the growth of B. papyrifera was significantly inhibited, specifically manifested as stunted plant growth and yellowing leaves.

(Chen et al. Comparative Physiological and Transcriptome Analysis Reveals the Potential Mechanism of Selenium Accumulation and Tolerance to Selenate Toxicity of Broussonetia Papyrifera. Tree Physiol 2022, 42, 2578–2595, doi:10.1093/treephys/tpac095.

Arı et al. Bioaccessibility and Bioavailability of Selenium Species in Se-Enriched Leeks (Allium Porrum) Cultivated by Hydroponically. Food Chemistry 2022, 372, 131314, doi:10.1016/j.foodchem.2021.131314.

Li et al. Effects of Selenate and Selenite on Selenium Accumulation and Speciation in Lettuce. Plant Physiol. Biochem. 2022, 192, 162–171, doi:10.1016/j.plaphy.2022.10.007.

Poblaciones et al. Effects of Selenate Application on Growth, Nutrient Bioaccumulation, and Bioactive Compounds in Broccoli (Brassica Oleracea Var. Italica L.). Horticulturae 2024, 10, 808, doi:10.3390/horticulturae10080808.)

  1. It is unclear why different designs of Se treatments were used for Broussonetia papyrifera and Arabidopsis thaliana (lines 117, 133).

Response: We apologize for the lack of explicit clarification in the original manuscript, as this difference was intentionally designed to align with the distinct biological characteristics of the two species, rather than arbitrary variation.

The two plant species differ drastically in growth habit, leaf structure, and Se metabolism efficiency. B. papyrifera has a longer growth cycle, thicker leaf cuticles (which reduce nutrient absorption efficiency), and larger biomass compared to herbaceous plants. To ensure sufficient Se uptake and translocation to tissues, we increased the total number of Se applications to 4 times (vs. 3 for Arabidopsis), with a 7-day interval between sprays. This prolonged, multi-dose regime allowed Se to gradually penetrate the thick cuticles and accumulate in leaves, avoiding insufficient Se levels due to low single-dose absorption. By contrast, as a model herbaceous plant, Arabidopsis has thin leaf cuticles (facilitating rapid nutrient absorption), a short life cycle (~5–6 weeks from seed to maturity), and high metabolic activity. Given its efficient selenium absorption and short growth cycle, three sprays (with a 7-day interval) are sufficient to achieve detectable selenium accumulation without inducing excessive stress.

To avoid confusion, we have added explanations of the species-specific in the "Materials and Methods" section of the revised manuscript.

  1. A shortcoming of the manuscript is the lack of the data on the effects of Se treatments on plants, at least regarding the simplest morphometric or physiological parameters. Without such data, it is difficult to assess the effects of the applied Se treatments on plants. Therefore, these data should be incorporated.

Response: Thank you for your comments. The research results of our team on the impact of selenium on Broussonetia papyrifera have been published in the journal Tree Physiology (Chen et al., 2022). We have supplemented the relevant content in the Introduction section:

Chen et al. found that exogenous Se treatment on B. papyrifera significantly increased the content of selenomethionine (the major organic Se form) in its leaves, while also elevating the contents of nutritional substances including soluble sugars, phenolic acids, and flavonoids, thereby improving the quality of B. papyrifera [6].

  1. It is necessary to check the spelling of the Latin names of the species in italics throughout the text (e.g., line 207).

Response: Thank you very much for your comments. According to the reviewer's suggestions, we have checked all the Latin names of species in the manuscript.

  1. The authors indicate that "Figure 8 also shows that the expression of three S-adenosylmethionine synthetase (SAM) genes and one methionine S methyltransferase (MMT) gene was moderately increased" (lines 464-466). However, this increase is insignificant in most cases except for SAM1 according to the data shown in Figure 8. It is important to correct the text (in Results, Discussion, etc.) according to the data of the statistical analysis.

Response: Thank you for your careful review and valuable comment. In the revised manuscript, we revised the descriptions of the relevant sections, including the Results and Discussion sections, to ensure that all textual descriptions are fully consistent with the statistical results of Figure 8.

Round 2

Reviewer 1 Report

Comments and Suggestions for Authors

The authors have made a number of text changes which would be useful to the reader.  Unfortunately they have not added additional data or experiments to address the major concerns which I raised.  Accordingly, I continue to feel this manuscript is too preliminary for publication in "plants".

Author Response

The authors have made a number of text changes which would be useful to the reader.  Unfortunately they have not added additional data or experiments to address the major concerns which I raised.  Accordingly, I continue to feel this manuscript is too preliminary for publication in "plants".

Response: Thank you.

Reviewer 3 Report

Comments and Suggestions for Authors

The manuscript has been revised.

Author Response

The manuscript has been revised.

Response: We greatly appreciate your insightful and helpful comments on our manuscript ,thank you very much!